# Mechanism of Bile Acid-Induced Programmed Cell Death and Drug Discovery against Cancer: A Review

**DOI:** 10.3390/ijms23137184

**Published:** 2022-06-28

**Authors:** Jung Yoon Jang, Eunok Im, Yung Hyun Choi, Nam Deuk Kim

**Affiliations:** 1Department of Pharmacy, College of Pharmacy, Research Institute for Drug Development, Pusan National University, Busan 46241, Korea; jungyoon486@pusan.ac.kr (J.Y.J.); eoim@pusan.ac.kr (E.I.); 2Department of Biochemistry, College of Korean Medicine, Dong-Eui University, Busan 47227, Korea; choiyh@deu.ac.kr

**Keywords:** bile acid, programmed cell death, apoptosis, autophagy, necroptosis, synthetic bile acid derivatives

## Abstract

Bile acids are major signaling molecules that play a significant role as emulsifiers in the digestion and absorption of dietary lipids. Bile acids are amphiphilic molecules produced by the reaction of enzymes with cholesterol as a substrate, and they are the primary metabolites of cholesterol in the body. Bile acids were initially considered as tumor promoters, but many studies have deemed them to be tumor suppressors. The tumor-suppressive effect of bile acids is associated with programmed cell death. Moreover, based on this fact, several synthetic bile acid derivatives have also been used to induce programmed cell death in several types of human cancers. This review comprehensively summarizes the literature related to bile acid-induced programmed cell death, such as apoptosis, autophagy, and necroptosis, and the status of drug development using synthetic bile acid derivatives against human cancers. We hope that this review will provide a reference for the future research and development of drugs against cancer.

## 1. Introduction

Bile acids are significant components of bile, accounting for approximately 85% of the solid ingredients in bile [1]. Bile acids are synthesized from cholesterol in the liver, and promote the absorption of fatty acids and cholesterol from dietary sources in the gastrointestinal tract. They also play an important role in maintaining signal processing and organizational homeostasis [2]. According to their chemical structures, different bile acids have distinct biological effects [3]. After being synthesized in the liver and discharged into bile ducts and digestive tracts, primary bile acids, such as cholic acid (CA) and chenodeoxycholic acid (CDCA), are metabolized by intestinal bacteria to produce secondary bile acids, such as deoxycholic acid (DCA) and lithocholic acid (LCA), and tertiary bile acids, such as ursodeoxycholic acid (UDCA) (Figure 1). Normally, more than 95% of bile acids are retained in the enterohepatic circulation, and the remaining 5% is excreted in feces [4,5] (Figure 2). In healthy individuals, the total amount of bile acids in enterohepatic circulation is estimated to be approximately 3 g, consisting of 35% CA, 40% CDCA, and 20% DCA [6,7]. Bile acid hydrophobicity depends not only on the number, location, and orientation of hydroxyl groups, but also on amidation at the C-24 position. The degree of hydrophobicity of the major bile acids is as follows: LCA > DCA > CDCA > CA > UDCA [8,9,10]. Bile acids conjugate with either glycine (~75%) or taurine (~25%) to form a more hydrophilic amidated form, which is dominated by glycine binding [11,12]. The conjugation of glycine or taurine makes bile acids non-toxic. They provide permeability through cell membranes and improve water solubility [2].

In the 1940s, bile acids were shown to cause cancer in rodents injected subcutaneously with the secondary bile acid DCA [13]. This resulted in malignant spindle cell tumors presenting with epitheliomas (benign growth or malignant carcinoma, classified according to the epithelial cell of origin). These early observations correlated with epidemiologic studies showing an association between bile acids and cancer, particularly colorectal cancer. Since then, several studies have shown that bile acids act as tumor promoters in various organs, such as the stomach, liver, esophagus, and colon [14,15,16,17]. However, in recent years, several studies have demonstrated that bile acids act as tumor suppressors, reducing the proliferation and migration of cancer cell types [18,19,20]. Moreover, several studies have reported that the bile acid/gut microbiome axis significantly reduces the classical in vitro cancer features of malignancies (cell invasion, clonogenicity, cell migration, and cell adhesion) [21]. In addition, several laboratories have synthesized novel bile acid derivatives to explore the tumor-suppressive function of bile acids. These derivatives were shown to induce programmed cell death in several cancer cells, and have tumor-suppressive effects [22,23,24,25,26,27,28,29,30,31,32]. 

Cell death plays a crucial role in embryonic development, maintains homeostasis, and removes damaged cells. Cell death can be classified as either programmed cell death or non-programmed cell death, depending on their signal dependence [33]. In 2018, the Nomenclature Committee on Cell Death (NCCD) listed several types of cell death in a molecule-oriented manner. The NCCD states that the fully physiological form of regulated cell death is commonly referred to as programmed cell death. Various types of programmed death have been described, including apoptosis (extrinsic and intrinsic), autophagy-dependent cell death, necroptosis, mitochondrial permeability transition-driven necrosis, ferroptosis, pyroptosis, parthanatos, entotic cell death, NETotic cell death, lysosome-dependent cell death, and immunogenic cell death [34]. 

In this review, we discuss type I cell death (apoptosis), type II cell death (autophagy-dependent), and programmed necrosis (necroptosis) among the programmed cell death types induced by natural bile acids and synthetic derivatives in cancer cells.

## 2. Role of Natural Bile Acids in Cancer

Bile acids are major signal molecules that play an important role as emulsifiers in the digestion and absorption of dietary lipids [21]. The role of bile acid in cancer (colon, esophageal, stomach, and intestines) is widely documented in the literature [35,36,37,38]. A combination of mechanisms appears to mediate bile-induced cancer in these organs with elevated bile acid levels, such as cell death, oxidative damage (ROS), epithelial proliferation, signal activation, and localized DNA instability [39]. Specific examples of cancer development by bile acids are as follows. Activation of the NF-κB pathway, a transcription factor involved in DNA transcription, the generation of inflammatory cytokines, and enhanced cell survival, occur in response to DCA exposure in the esophagus and colon [40,41,42]. Similarly, the expression of cyclooxygenase 2 (COX-2) and prostaglandin E2 is significantly promoted in pancreatic cancer cells (BxPC-3 and SU 86.86) in response to DCA and CDCA [43]. In patients with colorectal adenoma or carcinoma, the concentrations of DCA and LCA in the systemic circulation and the colon lumen as secondary bile acids are elevated [44]. UDCA is the most hydrophilic bile acid, while DCA and LCA are the least polar and most toxic bile acids that can contribute to the development of colorectal cancer [45]. 

Despite these observations, the major in vitro morphological features of cancer cells (e.g., cell infiltration, cell migration, cell adhesion, and cell survival) can be targeted by bile acids to inhibit the metastatic phenotype in some cancer models. Indeed, CDCA, DCA, and LCA have been shown to inhibit proliferation and induce the cell differentiation of leukemia HL60 cells via protein kinase C [46]. The proliferative capacities of the pancreatic cancer cells, MIA PaCa-2, PANC-1, and PGHAM-1, were studied in response to DCA. Reduced cell proliferation and structural changes in microvilli were observed in some cell lines, suggesting that DCA may limit pancreatic cancer cell proliferation and invasion in vitro [47]. Similarly, DCA has been shown to decrease the migration, invasion, and proliferation of MKN45 and SNU-216 gastric carcinoma cells [48]. In addition, UDCA has been shown to have antiproliferative and proapoptotic effects on M14 human melanoma cells. In M14 cells, UDCA activated the intrinsic apoptotic pathway and induced G2/M cell cycle arrest. At the same time, UDCA showed little toxicity to LO2 hepatocytes and HaCaT keratinocytes, which are normal human cell lines [49]. Studies performed on hepatocellular carcinoma and colitis-associated colon cancer have demonstrated that the anti-inflammatory activity of tauroursodeoxycholic acid (TUDCA) can be the basis for its anticancer potential. In carcinogen-induced liver dysfunction and hepatocellular carcinoma, TUDCA has shown chemopreventative properties by inhibiting endoplasmic reticulum stress (ER stress) and reducing hepatic inflammation [50]. In addition, in HCT-116 human colon cancer cells stimulated by tumor necrosis factor (TNF)-α, TUDCA significantly reduced the expression of IL-8 and IL-1α, and inhibited TNF-α-induced phosphorylation/degradation of IκBα, along with the inhibition of NF-κB DNA-binding activity [51]. These studies suggest that TUDCA may ameliorate tumorigenesis, primarily through the alleviation of NF-κB-mediated inflammatory responses. Bile acids such as DCA and CDCA not only promote the destabilization of hypoxia-inducible factor-1α (HIF-1α), an important transcriptional factor involved in the tumor hypoxic switch, but also inhibited pro-cancer phenotypes such as the invasion, migration, adhesion, and clonogenicity of DU-145 prostate cancer cells [52]. 

UDCA has long been considered a promising therapeutic agent for the treatment of colorectal and liver cancers. The chronic enrichment of a rodent diet with UDCA reduced DCA concentrations in the stool, and the incidence of benign and malignant tumors in Fischer 344 rats [53]. Thus, UDCA appears to have a potential chemotherapeutic effect in humans. In human phase III trials, UDCA administration was associated with a marked reduction in the recurrence of colorectal adenoma with hyperplastic dysplasia in 1285 patients, which is a major finding in people who are susceptible to invasive colorectal carcinoma [54]. In addition, UDCA application reduces the risk of colorectal cancer in patients with primary sclerosing cholangitis and ulcerative colitis [55], patients with primary biliary cirrhosis [56], and patients with chronic liver disease [57], and reduces colorectal cancer recurrence in patients after colorectal tumor removal [54].

The function of bile acid signaling in cancer progression is not a single event, and probiotics, aging, diet, drugs, and various form a bile acid/microbial axis that influences bile acid levels and profiles. An unbalanced approach to diet, junk food, excessive medicine/probiotic intake, and alcohol can increase bile acid levels and alter the bile acid profile to pathophysiological levels. This leads to cell membrane damage and DNA instability due to excessive ROS production. The downstream activation of inflammatory pathways and important regulatory factors (NF-κB, PKC, EGFR, etc.) can lead to or aggravate cancer cell proliferation in sensitive organs, such as the esophagus, colon, and stomach. In contrast, a healthy, balanced approach to diet, drug intake, and other factors may cause physiological levels of circulating bile acids [21].

## 3. Effect of Natural Bile Acids on Apoptosis

### 3.1. Apoptosis

The term apoptosis was first used by Kerr et al. [58] in 1972 to describe a morphologically striking type of cell death. Apoptosis is a process in which cells stop growing and dividing; instead, they do not spill content into the surrounding environment, but ultimately lead to controlled cell death. Apoptosis is also referred to as programmed cell death (or as ‘cellular suicide’) [59]. Apoptosis is a fundamental physiological process. Its dysregulation has been associated with various pathologies and diseases, including immune responses, drug toxicity, infections, tumors, and metabolic disorders [60]. Apoptosis is characterized by cell shrinkage, chromatin condensation, chromosomal DNA fragmentation, nuclear fragmentation, and cell membrane blebbing [61,62].

### 3.2. Types of Apoptosis

Apoptosis occurs mainly via the extrinsic death receptor pathway or the intrinsic mitochondrial pathway [63]. Intrinsic apoptotic pathways are initiated by a variety of intracellular stimuli, including growth factor deprivation, DNA damage, and oxidative stress [64]. This depends on the formation of a complex called the apoptosome consisting of pro-caspase-9, cytochrome *c*, and apoptotic protease activating factor 1 (Apaf-1). B-cell lymphoma-2 (Bcl-2) family members, such as Bcl-2 associated X protein (Bax), Bcl-2, Bcl-2 antagonist killer 1 (Bak), and B-cell lymphoma extra-large (Bcl-xL), regulate mitochondrial membrane permeability, thereby controlling the release of cytochrome *c* [63]. The Bcl-2 family can be divided into proapoptotic or antiapoptotic. The proteins of the proapoptotic class are, for example, Bad, Bcl-Xs, Bak, Bid, Bax, Bik, Bim, and Hrk, while those of the antiapoptotic class are Bcl-2, Bcl-XL, Bcl-W, Bfl-1, and myeloid leukemia 1 (Mcl-1) [65]. Cytochrome *c* forms an apoptosome in combination with Apaf-1, which recruits pro-caspase-9. In the apoptosome, caspase-9 is activated by autoproteolytic cleavage to initiate the caspase-processing cascade [66].

The extrinsic pathway of apoptosis is initiated by the interaction of a death ligand (e.g., TNF-α, TNF-related apoptosis-induced ligand (TRAIL), and Fas ligand (FasL)) with the death receptor of the TNF receptor superfamily. This interaction results in the assembly of a death-inducing signaling complex (DISC) consisting of the Fas-associated death domain (FADD) protein and pro-caspase-8/10. DISC then activates downstream effector caspases (caspase-3, 6, and 7) to induce cell death or cleave the Bcl-2 family member Bid to tBid to initiate the mitochondrial-mediated intrinsic apoptosis pathway [65]. Numerous factors have been reported to be involved in the regulation of apoptotic pathways, such as p53, NF-κB, and cellular inhibitors of apoptosis proteins (cIAPs). Many small molecules have been developed to target apoptotic pathways for cancer treatment [62]. 

### 3.3. Bile Acid-Related Apoptosis

The hydrophobicity of bile acids (in the order of LCA > DCA > CDCA > CA > UDCA) is closely related to the number and position of the hydroxyl groups attached to bile acids, and is a fundamental factor determining their biological activity [67]. However, the cytotoxicity of bile acids at the level of apoptosis induction does not always correlate with hydrophobicity. Apoptosis induction is dependent on bile acid concentration or conjugation status [68]. The molecular targets of bile acid-induced apoptosis are summarized (Table 1).

#### 3.3.1. Effects of Primary Bile Acids on Apoptosis

The synthesis of primary bile acids is divided into two biosynthetic pathways, which are defined as the classical and alternative pathways, also called neutral and acid pathways, respectively [94,95]. The classical pathway, responsible for approximately 90% of bile acid production under normal circumstances, is considered to be the major pathway for bile acid synthesis. Cholesterol 7α-hydroxylase (CYP7A1) initiates the classical pathway, is a rate-limiting enzyme of this pathway, and is located in the endoplasmic reticulum. It then passes through the sterol 12α-hydroxylase (CYP8B1) and the sterol 27-hydroxylase (CYP27A1) to form the primary bile acids CA and CDCA, respectively. The alternative pathway is initiated by CYP27A1, which converts cholesterol to 27-hydroxycholesterol via a hydroxylation reaction. 27-Hydroxycholesterol is continuously converted to CDCA instead of CA, with the participation of oxysterol 7α hydroxylase (CYP7B1). The alternative pathway is the secondary pathway of bile acid synthesis, which accounts for approximately 10% of bile acid production, and is generally considered active in pathological conditions [67,96,97]. After primary bile acid synthesis, taurine or glycine is conjugated to a 1:3 ratio via covalent modifications (known as bile salts) that improve solubility while reducing toxicity [98]. CA and CDCA are important primary bile acids synthesized in the human liver, and are conjugated with glycine or taurine for secretion into the bile [99]. 

Several studies have reported that CDCA, a primary bile acid, induces apoptosis in human cancer cells [69,70,71]. Juan et al. [69] studied the effects of DCA and CDCA treatment on BCS-TC2 human colon cancer cells, and found that both types of bile acids promoted apoptosis, and that the cell death effect was higher with CDCA. Bile acid-induced apoptosis causes oxidative stress, along with ROS generation. This effect leads to the loss of mitochondrial potential and the cytoplasmic release of proapoptotic factors, as confirmed by the activation of caspase-3 and -9. However, caspase-8 was not activated. This early apoptotic stage facilitates the cleavage of Bcl-2, allowing for Bax activation and pore formation in the mitochondrial membrane to amplify apoptotic signals. In addition, Shen et al. [71] showed that treatment of the lung adenocarcinoma A549 and H1650 cells with CDCA resulted in the induction of apoptosis and inhibition of cell proliferation, invasion, and migration. When A549 and H1650 cells were treated with CDCA, the expression of the mesenchymal markers N-cadherin and Snail decreased, and that of the epithelial marker E-cadherin increased. These results confirmed that CDCA inhibits the epithelial–mesenchymal transition, migration, and invasion. In addition, it was confirmed that CDCA inhibits integrin β1 and the phosphorylation of focal adhesion kinase (p-FAK) by inhibiting integrin α5. CDCA arrested growth and increased the levels of DNA damage-inducible 45 (GADD45), a downstream gene of p53. In addition, CDCA induced apoptosis by increasing the expression of p21, P2xm, Mcl-1, and Bax, which are genes related to p53 upregulation; and by decreasing the expression of insulin-like growth factor-binding protein 3 (IGFBP3) and Bcl-2, which are genes related to p53 downregulation. Through the evaluation of A549 cell xenografts in nude mice, it was confirmed that CDCA suppressed tumor volume and weight, and reduced integrin α5 and p-FAK levels. Therefore, they showed that CDCA regulates epithelial–mesenchymal transition, migration, invasion, and apoptosis in lung adenocarcinoma cells via the integrin α5β1/FAK/p53 axis.

Several studies have shown that glycochenodeoxycholic acid (GCDCA), which is glycine that is conjugated to CDCA, induces apoptosis in hepatocellular carcinoma cells [72,73,74]. Iizaka et al. [74] revealed the interaction between caspase-8 activation and ER stress in GCDCA-induced apoptosis in HepG2 hepatocellular carcinoma cells. They found that GCDCA-treated HepG2 cells showed increases in LDH leakage, cleaved caspase-3 proteins, cytochrome *c* release from the mitochondria, expression of the ER-resident molecular chaperone Bip mRNA, and the ER stress response-related transcription factor C/EBP homologous protein (CHOP) mRNA. In addition, GCDCA treatment increased the cleavage of BAP31, an integral membrane protein of the ER, and pretreatment with the caspase-8 inhibitor, Z-IETDFMK, inhibited an increase in caspase-8 and BAP31 cleavage. Thus, they demonstrated that activated caspase-8 promotes the ER stress response by cleaving BAP31 in GCDCA-induced apoptotic cells.

#### 3.3.2. Effects of Secondary and Tertiary Bile Acids on Apoptosis

The conjugated primary bile acids in the intestine are converted to the free primary bile acids CA and CDCA through the activity of bile salt hydrolases secreted by intestinal bacteria. The multi-stage 7α- dehydroxylation pathway catalyzes the dehydration of CA and CDCA to form secondary bile acids. CA is converted to DCA, and CDCA is converted to LCA and UDCA [67]. DCA, a secondary bile acid, has been shown to induce apoptosis in several human gastric cancer cells [41,75,76]. According to the results of Yang et al. [75], DCA induces apoptosis in BGC-823 gastric cancer cells through a p53-mediated pathway. When BGC-823 cells were treated with DCA, the expression of Bax and p53 proteins increased, and the expression of cyclin D1, Bcl-2, and cyclin-dependent kinase 2 (Cdk2) decreased. These results demonstrated that DCA inhibits cell growth and arrests cells at the G1 phase of the cell cycle. They also confirmed that DCA induces apoptosis that is associated with the disruption of the mitochondrial membrane potential. It was confirmed that the mitochondrial-dependent pathway was activated by an increase in the ratio of Bax:Bcl-2. Therefore, they verified that DCA induces apoptosis in gastric cancer cells through the activation of a mitochondria-dependent pathway involving p53. Several studies have also shown that DCA induces the apoptosis of colon cancer cells [69,77,78,79,80,81]. In a study by Yui et al. [78], Bax-knockout (Bax(−/−)) HCT-116 cells were treated with DCA to examine the role of Bax, a proapoptotic member of the Bcl-2 family, in apoptosis induction. As a result, the treatment of both Bax(−/−) and Bax(+/−) HCT-116 cells with DCA resulted in cytochrome *c* release and the activation of caspase-3, -8, and -9. These results demonstrated that Bax is not essential for DCA-induced apoptosis in HCT-116 cells.

LCA, the most hydrophobic secondary bile acid, has been reported to induce apoptosis in several studies [82,83,84,85,86]. According to a study by Luu et al. [82], when MCF-7 breast cancer cells were treated with LCA, the expression of proapoptotic p53 protein and antiapoptotic Bcl-2 protein decreased. In addition, LCA decreased Akt phosphorylation in MCF-7 cells, but not in MDA-MB-231 cells. They also showed that the LCA treatment of both breast cancer cell lines reduced the expression of sterol regulatory element-binding protein-1c (SREBP-1c), fatty acid synthase (FASN), and acetyl-CoA carboxylase (ACACA), which resulted in fewer lipid droplets compared to untreated control cells. Furthermore, it was confirmed that ERα expression was reduced when MCF-7 cells were treated with LCA. They demonstrated a potential therapeutic role of LCA in breast cancer cells by reversing the deregulation of lipid metabolism. Goldberg et al. [83] showed that LCA selectively induces apoptosis in androgen-dependent LNCaP and androgen-independent PC-3 prostate cancer cells. In LNCaP and PC-3 cells, LCA induced an extrinsic apoptosis pathway and activated caspase-3 and -8. It was also reported that LCA decreases the cleavage of Bid and Bax, downregulates Bcl-2, decreases mitochondrial membrane potential, and increased the activation of caspase-9. Furthermore, LCA has been reported to induce apoptosis in neuroblastoma cells [84,85]. Trah et al. [84] showed that the treatment of WT-CLS1 and SK-NEP1 nephroblastoma cells with LCA increased the levels of the G protein-coupled bile acid receptor (TGR5) without affecting the expression of NRF2, a downstream regulatory protein of the TGR5 pathway, and caused cytotoxicity due to caspase-3 and -7 activation.

Several studies have reported that UDCA, a tertiary bile acid with lower hydrophobicity, induces apoptosis in several cancer cell lines [18,20,49,87,88,89,90,91,92,93]. According to a study by Yu et al. [49], UDCA induces apoptosis in M14 and A375 human melanoma cell lines via the mitochondrial pathway. UDCA arrested cells at the G2/M phase of the cell cycle and reduced the expression of Cdk1 and cyclin B1. UDCA has been shown to induce the apoptosis of human melanoma M14 cells via ROS-induced mitochondrial-related pathways, resulting in increased p53, p21, Bax/Bcl-2 ratios, cytochrome *c* release, and Apaf-1, and cleaved caspase-3, -9, and PARP. In addition, treatment with Z-VAD-FMK (a caspase inhibitor) greatly reduced the rate of apoptosis. UDCA also reduced the expression of MMP-2 and MMP-9, which are related to cell migration. Yao et al. [87] reported that UDCA induced apoptosis in glioblastoma, A172, and LN229 cells. UDCA induced cell cycle arrest in the G1 phase; decreased the expression of Cdk2, Cdk4, Cdk6, cyclin D1, and pRb; and increased the expression of p21 and p53. They also showed that UDCA induced apoptosis in a caspase-independent manner. However, UDCA caused a decrease in mitochondrial membrane potential, and an increase in ROS. It was confirmed that the increase in ROS by UDCA was related to ERK, a member of the mitogen-activated protein kinase (MAPK) pathway. It was confirmed that the expression of CHOP, ATF6, and IRE1α, which are related to ER stress, was increased upon UDCA treatment. They also showed that combining UDCA with the proteasome inhibitor bortezomib can synergistically increase their levels by prolonging ER stress. Lee et al. [88] reported that UDCA induces apoptosis in DU145 prostate cancer cells. UDCA activated caspase 8, showing that UDCA-induced apoptosis is related to an extrinsic pathway. Furthermore, UDCA increased the expression of TRAIL, death receptor 4 (DR4), DR5, TRAIL, Bax, and cytochrome *c*, and downregulated the expression of Bcl-xL in DU145 cells. Thus, it was shown that UDCA induced the apoptosis of DU145 prostate cancer cells through the extrinsic and intrinsic pathways. In a study by Liu et al. [18], UDCA was shown to induce apoptosis in BEL7402 hepatocellular carcinoma xenografts in mice. UDCA suppressed tumor growth and increased DNA fragmentation.

In addition, the expression of Bax, Apaf-1, and cleaved caspase-3 and-9 was increased, and the expression of Bcl-2 was reduced. These results showed that UDCA inhibited the growth of hepatocellular carcinoma cells by inducing apoptosis. Lim et al. [91] showed that UDCA induced apoptosis in SNU601 gastric cancer cells through the MEK/ERK pathway. The reduction in UDCA-induced apoptosis using the MEK1 inhibitor PD98059 and the MEK1/2 inhibitor U0126 was demonstrated by an increase in apoptotic body formation, caspase-3, -6, and -8 activity, and PARP cleavage. U0126 reduced UDCA-induced TRAIL-R2/DR5 expression. Jung et al. [93] showed that UDCA induced apoptosis in FRO anaplastic thyroid cancer cells. UDCA increased the expression of proapoptotic (Bax, caspase-3, cytochrome *c*, and PARP) proteins, and inhibited the expression of antiapoptotic (Bcl-2) and angiogenic (TGF-β, VEGF, N-cadherin, and SIRT-1) proteins. In addition, UDCA treatment suppressed Akt and mammalian target of rapamycin (mTOR) phosphorylation. These results demonstrate that UDCA induces apoptosis and inhibits angiogenesis by regulating the Akt/mTOR signaling pathway.

## 4. Effect of Natural Bile Acids on Autophagy

### 4.1. Autophagy

The term autophagy is derived from the Greek “auto” (self) and “phagy” (to eat). Type II cell death refers to essential, regulated, and preserved catabolic processes that mediate the recycling and degradation of diverse cytoplasmic eukaryotic cell components [65,100]. Autophagy plays an important role in cancer, along with autophagy-associated (ATG) proteins [101]. Thus far, the dual role of autophagy in cancer progression and inhibition remains controversial. Autophagy plays a dynamic tumor suppressor or tumor-promoting role in various stages of cancer development. In early tumorigenesis, autophagy prevents tumor initiation and inhibits cancer progression via survival pathways and quality control mechanisms. However, as a tumor progresses, establishes a terminal stage, and is exposed to environmental stress, autophagy acts as a dynamic degradation and recycling system that provides the survival and growth of established tumors and promotes cancer aggression through metastasis. This indicates that the modulation of autophagy can be used as an effective interventional strategy for cancer treatment [102].

### 4.2. Types of Autophagy

Three types of autophagy have been identified based on the delivery method to the lysosome: macroautophagy, microautophagy, and chaperone-mediated autophagy (CMA) [103]. Macroautophagy, the most functional and characteristic form of autophagy, involves the formation of double-membrane autophagosomes, which remove damaged organelles or unwanted cellular components by delivery to lysosomes for degradation and recycling. Numerous reports have suggested that macroautophagy and macroautophagic cell death are antitumor responses [59,65]. In microautophagy, the cargo (organelles or cytoplasmic components) interacts directly and fuses with lysosomes. Microautophagy is more specific than macroautophagy, and transmits the signals of molecules that are present on the surface of damaged small organelles such as mitochondria and peroxisomes, causing specific fusion between lysosomes and these organelles. Depending on the organelle targeted, the generated autophagic vesicles are referred to by specific names. For example, for mitochondria, peroxisomes, lipids, and RNAs, the terms used are mitophagy, peroxophagy, lipophagy, and ribophagy, respectively [104]. Interestingly, CMA refers to the chaperone-dependent selection of soluble cytoplasmic proteins to be targeted to lysosomes; they are translocated across the lysosomal membrane for degradation. A unique feature of this type of autophagy is the direct shuttling of these proteins without requiring the selectivity of the degraded proteins and the formation of additional vesicles. The upregulation of CMA is associated with cancer cell survival and proliferation [105].

### 4.3. Bile Acid-Related Autophagy 

The molecular targets of bile acid-induced autophagy are summarized in Table 2. Most studies on bile acid-induced programmed cell death focus on apoptosis; however, there are several studies related to autophagy. Beclin-1 plays a central role in regulating autophagy in mammals [106]. The autophagic trigger increases Beclin-1 and binds to class III phosphatidylinositol 3-kinase (PI3KC3) to activate autophagosome formation and maturation. Beclin-1 acts as a tumor suppressor in mammals, and decreased Beclin-1 protein compared to normal tissue was found in breast cancer, colon cancer, cholangiocarcinoma, ovarian cancer, renal cell carcinoma, non-small cell lung cancer, and gastric cancer [107,108].

Roesly et al. [109] investigated the correlation between Beclin-1 expression and esophageal adenocarcinoma. Beclin-1 expression was high in normal esophageal epithelium and HET-1A cells (derived from normal squamous epithelium) but low in Barrett’s esophagus and esophageal adenocarcinoma cell lines (CP-A, CP-C, and OE33). Acute DCA exposure increased Beclin-1 expression and autophagy. However, chronic DCA exposure did not alter Beclin-1 levels or autophagy. Thus, they demonstrated that autophagy is initially initiated in response to DCA, but that chronic DCA exposure reduces Beclin-1 expression and autophagy resistance. Gafar et al. [110] reported that LCA, a secondary bile acid, induced autophagy in androgen-dependent LNCaP and androgen-independent PC-3 human prostate cancer cells. LCA induced the endoplasmic reticulum (ER) stress-related phosphorylation of eukaryotic initiation factor 2-alpha (p-eIF2α), CHOP, and c-Jun N-terminal kinases (p-JNK) in LNCaP and PC-3 cells. In addition, LCA induced the autophagy-associated protein ATG5 and the autophagy-related conversion of the microtubule-associated protein 1A/1B light chain 3B (LC3BI LC3BII) in PC-3 cells, but not in DU-145 cells. However, LCA induced mitochondrial dysfunction in the PC-3 and DU-145 cells. They demonstrated that LCA induced ER stress, autophagy, and mitochondrial dysfunction in human prostate cancer cells. Lim et al. [111] showed that UDCA induces autophagy in the cisplatin-resistant SNU601 gastric cancer sub-cell line (SNU601/R). In these cells, other anticancer drugs (etoposide, L-OHP, and rhTRAIL) affected the survival of resistant cells, but were sensitive to UDCA. The reduction in the viability of SNU601/R cells following treatment with UDCA was achieved through autophagy, whereas cell death in parental SNU601 cells (SNU601/WT) was caused through apoptosis. Previous studies have demonstrated that UDCA-induced apoptosis in gastric cancer cells is controlled by TRAIL-R2/DR5, and in SNU601/R cells, UDCA stimulation is regulated by TRAIL-R2/DR5, despite the absence of apoptosis. Therefore, these results demonstrate that UDCA has an advantage in overcoming drug resistance through apoptosis defects by inducing both apoptosis and autophagy cell death, depending on the intracellular signaling environment.

## 5. Effect of Natural Bile Acids on Necroptosis 

### 5.1. Necroptosis

Necroptosis, also known as programmed necrosis, is characterized by the activation of receptor-interacting protein kinases (RIPKs) via multiple signaling pathways [33]. Unlike apoptosis, necroptosis is a caspase-independent cell death [112]. Morphological changes, including organelle expansion, plasma membrane damage, and the release of cellular contents, can lead to the development of secondary inflammation [113,114,115,116]. RIPKs are activated when recruited by diverse cell surface receptors, such as death receptors (DRs), toll-like receptors (TLRs), and T-cell receptors (TCR), into a macromolecular complex [117,118]. The key to the necroptosis mechanism is the regulation of the formation of the necrosome, a complex that is composed of receptor-interacting serine/threonine-protein kinase 1 (RIPK1), RIPK3, and mixed lineage kinase domain-like (MLKL) [119,120,121]. RIPK3 induces MLKL oligomerization by further activating the downstream molecule MLKL via phosphorylation [122,123]. Oligomerized MLKL inserts into and permeates the cell membrane, eventually leading to cell death [124]. RIPK3-dependent necroptosis is induced by a DNA-dependent interferon activator (DAI) modulator, a cytosolic DNA sensor, following the presence of double-stranded viral DNA or viral infection [125]. In addition to its role in inducing cell death, necroptosis can induce an adaptive immune response by inducing the secretion of pro-inflammatory cytokines [126]. Necroptosis is associated with the development of pathologies characterized by unwarranted cell loss and inflammatory components [127].

### 5.2. Necroptosis in Cancer

Necroptosis has both pro-cancer and anticancer effects [100,128]. The dual effect of advancing and decreasing tumor growth has been observed in diverse cancer types [129,130,131,132]. As an unsafe form of cell death that occurs in cells in which apoptosis is not induced, necroptosis can prevent tumor development. Necroptosis can induce an inflammatory response and is known to promote cancer metastasis and immunosuppression [131,133]. Apoptosis in cancer cells has been induced to remove malignant cells [134]. However, the deregulation of apoptosis signals in cancer, especially the activation of anti-apoptosis systems, allows cancer cells to escape this program and induces uncontrolled cell proliferation and tumor survival. Therefore, tumor necroptosis, which is caspase-independent cell death, has great therapeutic potential for cancer treatment. Various natural compounds, chemotherapeutic agents, and classical necroptosis-inducing drugs have been found to induce the MLKL-mediated necroptosis of cancer cells [135]. For example, the natural compound shikonin and its analogs have been reported to cause necroptosis in myeloma and glioma cells [136,137]. Chemotherapeutic drugs, including 5-FU, etoposide, and cisplatin, can induce tumor cell necroptosis when caspase activity is prevented [138,139,140]. The necroptosis inducer second mitochondrial activator of caspases (Smac) mimetic BV6, can antagonize the inhibitor of apoptosis protein (IAP) and cause the necroptotic death of tumor cells, and thus is a potential alternative strategy for anticancer therapy [141,142]. Cekay et al. [143] demonstrated that the synergistic interaction of interferon-γ (IFN-γ) with BV6 induces IFN-regulatory factor 1 (IRF-1)-dependent necroptosis in diverse types of apoptosis-resistant cancer cells when caspase activity is blocked. In pancreatic cancer and acute myeloid leukemia cells, BV6 induces TNF-α production and the formation of necrosomes [141,144].

### 5.3. Bile Acid-Related Necroptosis

Since necroptosis has only recently been discovered as a new type of programmed cell death, studies on bile acids are scarce. Therefore, in addition to cancer, bile acid-related necroptosis associated with cholestasis and pancreatitis was also reviewed. The molecular targets of bile acid-induced necroptosis are summarized in Table 3. In necroptosis, the formation of the necroptosis-defining necrosome requires the phosphorylation of RIPK1 and the activation of MLKL, which result in pore formation and cell death [145,146,147]. The first step in necroptosis induction is shared with the apoptotic pathway. RIPK1 has been reported to be recruited to active cytokine receptors, such as tumor necrosis factor receptor type 1 (TNFR1), which is phosphorylated (p-RIPK1) in a multiprotein complex also known as complex I [148,149]. p-RIPK1 activates caspase-8, which induces apoptosis by interacting with Fas-associated death domain proteins. When caspase-8 action is inhibited, RIPK3 is phosphorylated, initiating the necroptosis pathway [150]. p-RIPK3 binds to MLKL in a multiprotein complex, triggering its phosphorylation (p-MLKL). The pRIPK3/pMLKL complex is incorporated into the membrane, where several p-MLKL proteins are oligomerized and form pores, finally leading to cell degradation [151]. Hoff et al. [152] reported the regulatory mechanism of RIPK3, an important molecule in necroptosis, following the treatment of hepatocytes and HepG2 hepatocellular carcinoma cells with bile acids. They confirmed the expression of RIPK1, RIPK3, and MLKL, which are important molecules for necroptosis, in HepG2 hepatocellular carcinoma cells, primary human hepatocytes (pHep), and primary human macrophages (pMФ). RIPK3 expression was absent in hepatocytes, whereas RIPK1 and MLKL were highly expressed in the three cell types HepG2, pHep, and pMФ. The treatment of HepG2 cells overexpressing an N-terminal FLAG-tagged human RIPK3 construct with the unconjugated hydrophilic bile acids CA and UDCA upregulated RIPK3-FLAG phosphorylation and RIPK3-FLAG levels. In addition, CDCA had a minor effect on RIPK3-FLAG phosphorylation without affecting RIPK3-FLAG levels. By contrast, LCA reduced RIPK3-FLAG phosphorylation and levels. TCA, GCA, taurine-conjugated CDCA (TCDCA), and glycine-conjugated CDCA (GCDCA) reduced the stimulatory effects of CA and UDCA on RIPK3-FLAG phosphorylation but did not prevent them. Interestingly, TCDCA reduced the levels and phosphorylation of RIPK3-FLAG without indications of toxicity; however, GCDCA had no effect. The effects of GLCA and taurine-conjugated LCA (TLCA) were identical to that of the unconjugated form. The inflammatory cytokine IL-8 increased CA-induced RIPK3-FLAG activation in HepG2 cells, with the most potent RIPK3-FLAG activation and phosphorylation at the protein level. In addition, only the hydrophobic bile acid CDCA significantly induced IL-8 expression. IL-8 secretion can be controlled by the c-Jun N-terminal kinases (JNK) signaling pathway [153,154]. Stimulation with hydrophilic bile acids (CA and UDCA) further enhanced JNK phosphorylation. These results show that RIPK3 expression and phosphorylation cause necroptosis, which, in turn, led to the regulation of IL-8 secretion by JNK.

Afonso et al. [155] showed that necroptosis was induced in a study of bile acid-related cholestasis. Primary biliary cholangitis (PBC) patients are chronic cholestatic liver disease characterized by the destruction of small intrahepatic bile ducts [156]. GCDCA is a major component of human serum and bile during cholestasis [157]. These studies showed increased RIPK3 expression and MLKL phosphorylation in liver samples from human PBC patients. Bile duct ligation (BDL) is a surgical model for serious obstructive cholestasis, resulting in significant jaundice and hepatocellular damage [158]. They found that the mRNA and protein expression of RIPK3 and MLKL, and MLKL phosphorylation were strongly increased in the liver of BDL mice. RIP1 mRNA levels were not altered, but RIPK1 protein levels were also increased in whole liver cell lysates from BDL mice. These results suggest that targeting necroptosis may represent a therapeutic strategy for acute cholestasis.

Zhou et al. [159] reported that bile acid induces necroptosis in chronic pancreatitis. Bile acid has also been shown to induce acinar cell death through decreased mitochondrial membrane potential, increased reactive oxygen species and energy depletion [160]. All of these are known to promote acinar cell apoptosis and necrosis in pancreatitis. The most abundant primary bile acids in humans are GCDCA and TCA [159]. Therefore, they confirmed that pancreatic cell lines exposed to GCDCA and TCA increased the expression of the nuclear bile acid receptor known as the farnesoid X receptor (FXR), and decreased the expression of the essential autophagy-associated protein ATG7. Bile acid was also increased in pancreatic tissue from patients with human chronic pancreatitis, which correlated with increased FXR, and ATG7 expression was associated with locally reduced autophagic activity. These results demonstrated a cascade of events in which the local accumulation of bile acid signals through FXR inhibits autophagy in pancreatic acinar cells, thereby triggering acinar cell apoptosis and necroptosis.

## 6. Cell Death Mechanism of Synthetic Bile Acid Derivatives in Cancer

Several synthetic bile acid derivatives have been reported to induce programmed cell death (Table 4). The structures of synthetic bile acid derivatives are shown in Figure 3.

First, many researchers have shown that the UDCA derivatives HS-1030 and HS-1183, and the CDCA derivatives HS-1199 and HS-1200 induced apoptosis in various human cancer cells [22,161,162,163,164,165,166,167,168,169,170,171]. HS-1030 is a form of UDCA conjugated with glycine methyl esters (Figure 3A,B). HS-1183 is a form of UDCA conjugated with L-phenyl alanine benzyl ester. HS-1199 is a form of CDCA conjugated with L-phenyl alanine benzyl ester. HS-1200 is a form of CDCA conjugated with β-alanine benzyl ester [22,172]. It has been shown that these derivatives inhibit the growth of several human cancer cells through apoptosis. Of the four synthetic bile acid derivatives tested, HS-1199 and HS-1200 significantly increased apoptosis, whereas HS-1030 and HS-1183 slightly increased apoptosis. Among these, HS-1200 had the strongest effect [22,167].

The tumor suppressor gene *TP53* is mutated in approximately 50% of all human cancers. In addition to its role in tumor suppression, p53 plays an important role in the response to many anticancer drugs, especially in malignant and unmodified cells, causing DNA damage. p53 forms homodimers that directly regulate approximately 500 target genes, thus controlling a wide range of cellular processes, including the cell cycle, cell senescence, DNA repair, metabolic adaptation, and cell death [173]. Several laboratories have tested synthetic bile acid derivatives in MCF-7 human breast cancer cells (wild-type p53), MDA-MB-231 (mutant p53), PC-3 human prostate cancer cells (null-type p53), and HT-29 human colon cancer cells (mutant p53). They found that bile acid derivatives arrested cell cycle progression in the G1 phase, with increased levels of the Cdk inhibitor, p21^WAF1/CIP1^, and induced apoptosis regardless of p53 status [22,163,164].

Apoptosis is mediated by a caspase (cysteine aspartyl-specific protease) that cleaves target proteins [174]. Caspase protease activity cleaves hundreds of diverse proteins and is essential for apoptosis [175]. There are four initiator caspases (caspases-2, -8, -9, and 10) and three executioner caspases (caspases -3, -6, and -7) [176]. Executioner caspases cleave target proteins, ultimately leading to cell death. This pathway is highly regulated and occurs only following apoptotic signaling. Treatment with synthetic bile acid derivatives (HS-1183, HS-1199, and HS-1200) reduced the levels of pro-caspase-3 and caspase-8 in Jurkat human leukemic T cells [161]. In addition, the levels of pro-caspase-3 were reduced in SNU-1 stomach cancer cells [165,166] and malignant glioblastoma cells (U-118MG, U-87MG, T98G, and U-373MG) [167]. Furthermore, HS-1200 decreased the levels of pro-caspase-3 and pro-caspase-7 in KAT-18 thyroid carcinoma cells [171]. Caspase-mediated signal transduction may contribute to synthetic bile acid (HS-1183, HS-1199, and HS-1200)-mediated apoptosis.

Intrinsic pathways of apoptosis are specifically controlled by the B-cell lymphoma-2 (Bcl-2) protein family, which contains proapoptotic effector proteins, proapoptotic BH3-only proteins, and antiapoptotic Bcl-2 proteins [175]. Antiapoptotic Bcl-2 proteins hinder apoptosis by inhibiting the proapoptotic Bcl-2 proteins BAX and BAK [177]. In MCF-7 cells treated with synthetic bile acid derivatives (HS-1183, HS-1199, and HS-1200), Bcl-2 expression was not significantly altered, but Bax expression was significantly increased [22,163]. In contrast, Bax expression levels were increased and Bcl-2 expression levels were significantly decreased in MDA-MB-231 and HCT-116 cells (wild type, null-type p21, and null-type p53) [163].

Early growth response 1 (Egr-1), also known as ZIF268, Krox24, NGFI-A, or TIS8, is a zinc-finger transcription factor that regulates transcription through a GC-rich consensus sequence of 5′-GCG(T/G)GGGCG-3′ [178]. Egr-1 is known as an instantaneous early response gene due to its fast kinetics of induction by various signals, including growth factors, cytokines, and stress DNA damage [179]. Egr-1 may be involved in cell proliferation, cell differentiation, and apoptosis. In cancer, Egr-1 can act as a tumor suppressor by promoting apoptosis in response to stress and DNA damage [180]. At present, there are many potential applications of the Egr-1 gene in cancer therapy. Previous studies have reported that bile acids upregulate Egr-1 in gastric cancer cells through the MAPK signaling pathway [181]. Another study demonstrated that the bile acid DCA increased Egr-1 protein levels in primary mouse hepatocytes. This suggests that the upregulation of Egr-1 in the liver during cholestasis could be bile acid-dependent [182]. Park et al. [169] also found that treatment with the synthetic bile acid HS-1200 significantly induced Egr-1 expression at an early stage. The protein expression of p53, p21^WAF1/CIP1^, p27^KIP1^, and COX-2 was downregulated by silencing Egr-1. Park et al. [169] found that Egr-1 plays an important role as a gene regulator in HS-1200-treated hepatocellular carcinoma cells, and is involved in important cellular processes such as apoptosis and cell cycle regulation.

JNK, also known as the stress-activating protein kinase of the MAPK family, has been implicated in many cellular events, including responses to various stress signals, apoptosis, and autophagy [183]. A large body of evidence suggests that bile acids can regulate cell growth and induce apoptosis by activating the JNK/activator protein-1 (AP-1) signaling pathway [184]. CDCA, LCA, and UDCA induce the activation of AP-1 in HT-29 and HCT-116 cells [185]. In addition, CDCA modifies AP-1 activity in hepatic stellate cells [186]. In addition, the induction of AP-1 by bile acids is known to require the activation of both ERK and PKC [187]. Recently, bile acids were shown to upregulate death receptor 5/TRAIL-receptor 2 expression via a JNK-dependent pathway, including Sp1 [188]. Im et al. [162] found that HS-1199 and HS-1200 significantly upregulate JNK phosphorylation. In addition, HS-1183, which induces less apoptosis in SiHa cells than HS-1199 and HS-1200, was also found to have a lesser effect on JNK phosphorylation.

The NF-κB protein is maintained in the cytoplasm in an inactive state by an inhibitory subunit called IκB. NF-kB consists of a DNA-binding domain (p50) and a transactivating domain (p65); it may enhance or promote gene induction and apoptosis [189]. Im et al. [162] showed that the treatment of human cervical cancer cells with HS-1200 reduced the levels of p65, p50, and IκB-α in a dose-dependent manner. These results confirmed that HS-1200 temporarily increased NF-kB activity, and that the translocation of the NF-kB active complex to the nucleus is involved in the regulation of transcription of other apoptosis-related genes in human cervical cancer cells.

Second, Katona et al. [23] found that the synthetic enantiomers of lithocholic acid (ent-LCA), chenodeoxycholic acid (ent-CDCA), and deoxycholic acid (ent-DCA) induced toxicity and apoptosis in HT-29 and HCT-116 colorectal cancer cells (Figure 3C). Native bile acids induced more apoptosis and cleavage of capase-3 and -9 compared to enantiomeric bile acids. However, natural and enantiomeric bile acids had similar effects on cell proliferation. Among them, LCA- and ent-LCA-mediated apoptosis were prevented by both the pan-caspase and selective caspase-8 inhibitors, whereas selective caspase-2 inhibitors provided no protection. In addition, LCA and ent-LCA increased CD95 localization in the plasma membranes. Bile acid-mediated caspase-8 activation in hepatocytes is induced by CD95 oligomerization and translocation to the cell membrane [92]. This showed that LCA and ent-LCA induced apoptosis selectively through CD95 activation, which induced pro-caspase-8 cleavage due to increased ROS production. 

Third, Agarwal et al. [24] showed that the bile acid-added triazolyl aryl ketones 6af and 6cf induced apoptosis in MCF-7 breast carcinoma cells (Figure 3D). In particular, compound 6cf induced apoptosis by 46.09% in MCF-7 cells, and compound 6af induced apoptosis by 33.89%, showing that 6cf was more effective in inducing apoptosis. 

Fourth, Melloni et al. [25] showed that CDC-PTX and UDC-PTX combined with paclitaxel (PTX), an anticancer drug, induced apoptosis in HL60 and NB4 acute promyelocytic leukemia cells through a high-yield condensation reaction of CDCA and UDCA (Figure 3E). It was also shown that CDC-PTX and UDC-PTX RKO induced apoptosis in HCT-116 colon cancer cells. In particular, in all four cell lines (HL60 and NB4 human leukemic cell lines, RKO, and HCT-116 colon cancer cells), CDC-PTX induced more apoptosis than UDC-PTX. In addition, Pacific Blue (PB)-conjugated derivatives of CDC-PTX and UDC-PTX (CDC-PTX-PB and UDC-PTX-PB) were prepared through multi-step synthesis to evaluate their ability to enter tumor cells. CDC-PTX-PB showed a greater ability to cross the plasma membrane than UDC-PTX-PB.

Fifth, Brossard et al. [26] showed that 7b, a new piperazinyl bile acid derivative, induced apoptosis in KMS-11 multiple myeloma and HCT-116 colon cancer cells (Figure 3F). Moreover, the apoptosis rate was higher in KMS-11 multiple myeloma cells than in HCT-116 colon cancer cells. Compound 7b was also shown to induce DNA fragmentation, a characteristic of apoptosis, in KMS-11 cells.

Sixth, Singh et al. [27] showed that four cationic bile acid-based facial amphiphilic substances, LCA-TMA_1_, CDCA-TMA_2_, DCA-TMA_2_, and CA-TMA_3_ characterized by trimethyl ammonium head groups, induced apoptosis in HCT-116 and DLD-1 colon cancer cells (Figure 3G). LCA-TMA_1_ induced the highest level of apoptosis in both cells.

Seventh, Kihel et al. [28] synthesized six novel bile acid (LCA and CDCA)-substituted piperazine conjugates lysocholic acids and chenodeoxycholic acid piperazinylcarboxamides, and showed that compound IIIb caused apoptosis in KMS-11 multiple myeloma cells (Figure 3H). Of the six compounds, IIIb showed the best apoptotic activity in KMS-11 multiple myeloma cells. This revealed that apoptosis is involved in Mcl-1 and PARP-1 cleavage, the inhibition of NF-κB signaling, and DNA fragmentation.

Eighth, Singh et al. [29] synthesized cationic amphiphilic materials with different cationic charge head group characteristics using LCA. Among them, it was confirmed that the lithocholic acid-based amphiphilic substance carrying the piperidine head group (LCA-PIP_1_) was 10 times more cytotoxic to colorectal cancer cells than the precursor (Figure 3I). This confirmed that LCA-PIP_1_ induced greater levels of apoptosis in HCT-116 colorectal cancer cells compared to LCA. LCA-PIP_1_ induced sub-G0 arrest and the cleavage of caspase-3, -7, and -8. These effects of LCA-PIP_1_ were also confirmed in a tumor xenograft model of HCT-116 cells; tumor volume was reduced by up to 75%.

Ninth, Sreekanth et al. [30] synthesized a bile acid conjugate of tamoxifen using three bile acids LCA, DCA, and CA (Figure 3J). Among them, the free amine headgroup-based CA-tamoxifen conjugate (CA-Tam_3_-Am) was shown to be the most potent anticancer conjugate in breast cancer cells compared to tamoxifen, and it induced apoptosis. CA-Tam_3_-Am induced more apoptosis than tamoxifen in 4T1, MCF-7, T47D, and MDA-MB 231 breast cancer cells, and showed cell arrest at the G0 phase. In addition, the treatment of MCF7 cells, which are estrogen receptor-positive, with CA-Tam_3_-Am induced apoptosis through the intrinsic and extrinsic pathways, whereas treatment of MDA-MB-231 cells, which are estrogen receptor-negative, resulted in apoptosis via the intrinsic pathway. 

Tenth, Tang et al. [31] showed that norUDCA, a side-shortened C23 homologue of UDCA, induces autophagy in HTOZ cervical cancer cells (Figure 3K). Alpha-1 antitrypsin (α1AT) deficiency is a genetic disorder that causes the accumulation of the α1AT mutant Z (α1ATZ) protein in the ER of hepatocytes, leading to chronic liver damage, liver fibrosis, and hepatocellular carcinoma [190]. NorUDCA inhibited the accumulation of α1ATZ through the autophagy-mediated degradation of α1ATZ in HTOZ cells. They showed that AMPK activation is required for norUDCA-induced autophagy and α1ATZ degradation. Furthermore, they demonstrated that mTOR/ULK1 was involved in norUDCA-induced AMPK activation and autophagy in HTOZ cells.

Eleventh, Markov et al. [32] showed that compound 9, among a series of novel DCA derivatives containing an aliphatic diamine and aminoalcohol or morpholine moiety at the C3 position, induces apoptosis and autophagy in HuTu-80 duodenal carcinoma cells (Figure 3L). They showed that compound 9 causes ROS-dependent cell death by activating the intrinsic caspase-dependent pathway of apoptosis and cell-destructive autophagy in HuTu-80 duodenal carcinoma cells.

## 7. Conclusions

This article reviews the mechanisms of programmed cell death in terms of apoptosis, autophagy, and necroptosis following treatment with natural bile acids and synthetic bile acid derivatives as tumor suppressors in cancer. Have bile acids a pro-cancer or an anticancer activity? This has long been a controversial topic. Much of the literature suggests an association between bile acids and some cancers, reinforcing the “pro-cancer” component of bile acids. However, some bile acids may induce an anticancer phenotype in cells that have already undergone early malignant transformation. This phenomenon is related to the amphiphilic structure of bile acids and the additional target pathways that are not triggered at physiological concentrations. In addition, the relationship between bile acids and cancer may be affected by the bile acid/gut microbiome axis. Synthesized bile acid derivatives induce programmed cell death in several human cancer cell lines. Thus, these new bile acid derivatives may be promising chemicals that specifically target a variety of cancer cells by inducing programmed cell death. These can be important lead compounds for the development of novel anticancer agents based on the structure of bile acids.

## Figures and Tables

**Figure 1 ijms-23-07184-f001:**
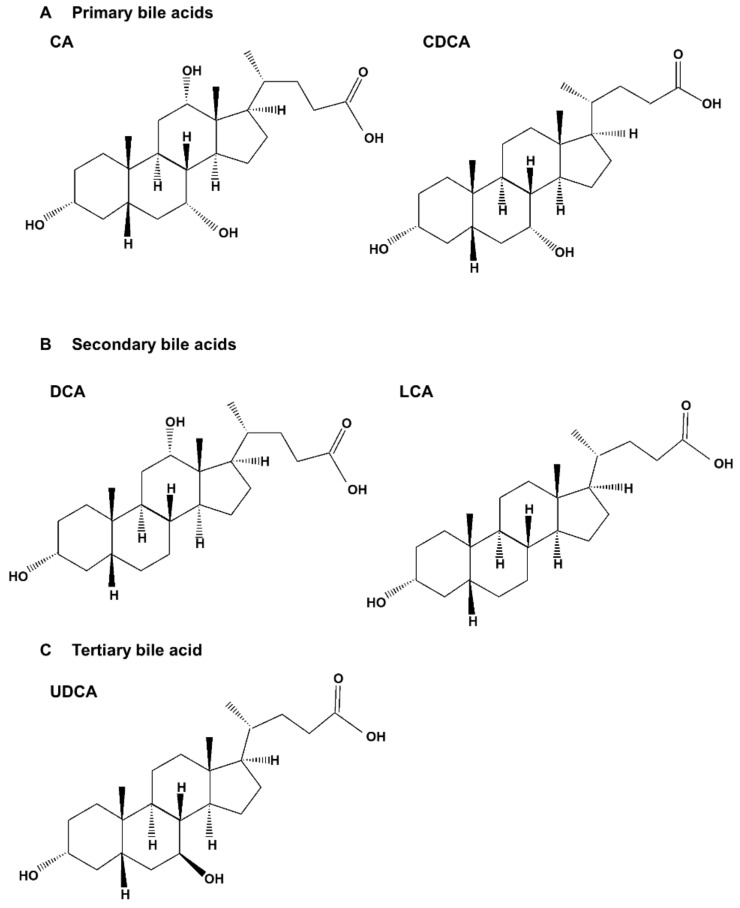
Molecular structures of bile acids. (**A**) Molecular structures of primary bile acids such as cholic acid (CA) and chenodeoxycholic acid (CDCA). (**B**) Molecular structures of secondary bile acids such as deoxycholic acid (DCA) and lithocholic acid (LCA), and (**C**) tertiary bile acids such as ursodeoxycholic acid (UDCA).

**Figure 2 ijms-23-07184-f002:**
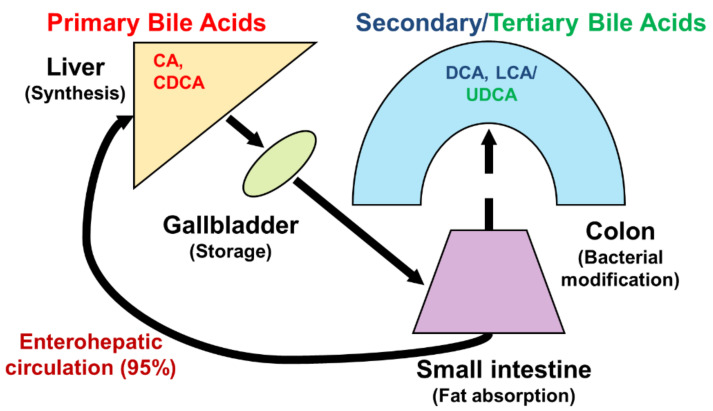
Bile acid circulation. Primary bile acids are synthesized from cholesterol in the liver. Then, they bind to glycine or taurine, are discharged into the bile, and are stored in the gallbladder. After a meal, conjugated bile acids are released into the intestinal tract to facilitate the digestion of dietary lipids and fat-soluble vitamins. Bile acids are then efficiently reabsorbed in the ileum, and most (95%) are transported back to the liver via the hepatic portal vein, where they are re-secreted in the bile to prepare for new circulation. This process is called enterohepatic circulation. In the large intestine, some bile acids are deconjugated to free bile acids by bacterial bile salt hydrolases, and are converted to secondary and tertiary bile acids. They are then reabsorbed into colonocytes, returned to the liver for detoxification, and recycled. Only a small amount (approximately 5%) of bile acids is lost through the feces.

**Figure 3 ijms-23-07184-f003:**
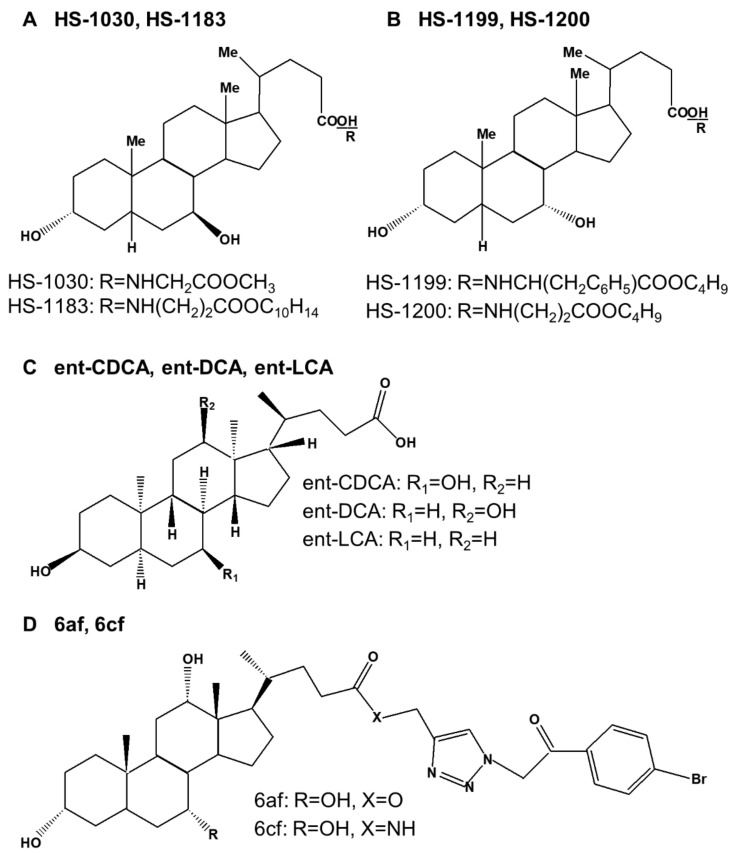
Structures of synthetic bile acid derivatives. CA, cholic acid; CA-TMA_3_, cholic acid based amphiphile; CA-Tam_3_-Am, cholic acid−tamoxifen conjugate; CDC-PTX, chenodeoxycholic-paclitaxel hybrid; CDCA, chenodeoxycholic acid; CDCA-TMA_2_, chenodeoxycholic acid based amphiphiles; compound IIIb, chenodeoxycholic acid-substituted piperazine conjugate; compound 9, chenodeoxycholic acid derivative; ent-CDCA, enantiomers of chenodeoxycholic acid; ent-DCA, enantiomers of deoxycholic acid; ent-LCA, enantiomers of lithocholic acid; DCA, deoxycholic acid; DCA-TMA_2_, deoxycholic acid based amphiphiles; HS-1030 and HS-1183, ursodeoxycholic acid derivatives; HS-1199 and HS-1200, chenodeoxycholic acid derivatives; LCA, lithocholic acid; LCA-PIP_1_, lithocholic acid–piperidine; LCA-TMA_1_, lithocholic acid based amphiphile; norUDCA, nor-ursodeoxycholic acid; UDCA, ursodeoxycholic acid; UDC-PTX, ursodeoxycholic-paclitaxel hybrid; 6af and 6cf; bile acid-added triazolyl aryl ketones; 7b, piperazinyl bile acid derivative.

**Table 1 ijms-23-07184-t001:** Molecular targets of bile acid-induced apoptosis.

Types	Target Molecules		Model(s)	Refs.
	Up-Regulation	Down-Regulation		
**CDCA**	mitochondrial transition permeability, ROS, caspase-3 and -9, cleavage of Bcl-2, Bax	ΔΨm	Colon cancer cells (BCS-TC2)	[69]
	cleavage of PARP, mitochondrial depolarization, Cyt *c* (cytosolic)		Hepatocellular carcinoma cells (HepG2)	[70]
	E-cadherin, p53, p21, Bax, GADD45, P2xm, Mcl-1	N-cadherin, Snail, integrin α5, integrin β1, p-FAK, IGFBP3, Bcl-2	Lung cancer cells (A549, H1650), xenograft (A549)	[71]
**GCDCA**	Cyt *c*, DR5, TNF-R1, cleaved caspase-3, -7, -8, and BAP31, AP-1, p-JNK, p-p38, Bax, Bip, CHOP, LDH	DR6	Hepatocellular carcinoma cells (HepG2, HepG2-Ntcp, HuH-BAT)	[72,73,74]
**DCA**	NF-κB (nuclear), caspase-3, -6, and -9, cleavage of PARP and PKC ε, PKC β1, ratio of Bax to Bcl-2, p53, cyclin D1	Bcl-2, cyclin D1, Cdk2, ΔΨm	Gastric cancer cells (AGS, BGC-823, SGC-7901)	[41,75,76]
	p-ERK, p-p38, p-Elk-1, p-ATF2, Cyt *c* release, caspase-3, -8, and -9, cleavage of PARP, ROS, p-p38, p-ERK1/2, cleavage of Bcl-2, Bax	c-Myc, ΔΨm, Bid	Colon cancer cells (HCT-116, BCS-TC2, HT-29)	[69,77,78,79,80,81]
**LCA**	p53	Bcl-2, p-Akt, SREBP-1c, FASN, ACACA, ERα	Breast cancer cells (MCF-7, MDA-MB-231)	[82]
	caspase-3, -8, and -9 activity, cleavage of PARP, Bid, Bax	Bcl-2, ΔΨm	Prostate cancer cells (PC-3, LNCaP)	[83]
	TGR5, caspase-3, -6, -7, -8, and-9 activity		Neuroblastoma cells (WT-CLS1, SK-NEP1, BE(2)-m17, SK-n-SH, SK-n-MCIXC, Lan-1)	[84,85]
	ROS	ΔΨm	Hepatocellular carcinoma cells (HepG2)	[86]
**UDCA**	ROS, cleaved caspase-3, -9, and PARP-1, Bax/Bcl-2 ratio, Apaf-1, p21, p53, Cyt *c*	ΔΨm, Cdk1, cyclin B1, Bcl-2, MMP-2 and -9,	Melanoma cells (M14 and A375)	[49]
	ROS, Bip, IRE1α, ATF4, ATF6, p-PERK, CHOP, p21, p53, p-ERK	ΔΨm, Cdk2, Cdk4, Cdk6, pRb, cyclin D1, RIP3, Bcl-2	Glioblastoma multiforme cells (A172, LN229)	[87]
	TRAIL, DR4, DR5, Bax, Cyt *c*, cleavage of PARP	Bcl-xL, pro-caspase-3 and -8	Prostate cancer cells (DU145)	[88]
	Bax, Samc, caspase-2, -3, -8, and -9, Apaf-1	Bcl-2, Livin	Hepatocellular carcinoma cells (HepG2)	[89,90]
	Bax, Apaf-1, cleavage of caspase-3 and -9, Cyt *c* (cytosolic)	Bcl-2, Cyt *c* (mitochondrial)	Hepatocellular carcinoma xenografts (BEL7402)	[18]
	p-ERK1/2, p-MEK1/2, caspase-3, -6, and -8, cleavage of PARP, DR5, TRAIL, ROS, PKCδ		Gastric cancer cells (SNU601, SNU638)	[91,92]
	DR5		Gastric cancer cell xenografts (SNU601)	[92]
	Bax, caspase-3, Cyt *c*, PARP	Bcl-2, TGF-β, VEGF, N-cadherin, SIRT-1, p-Akt, p-mTOR	Anaplastic thyroid cancer (FRO)	[93]
	caspase-3, -8, and -9, Bax, Fas, FasL, TRAIL, DR4, DR5, IκB-α	Bcl-2, Bcl-xL, XIAP, cIAP-1, cIAP-2, survival, NF-κB	Oral squamous carcinoma cells (HSC-3)	[20]

ACACA, acetyl-CoA carboxylase; AIF, apoptosis-inducing factor; AP-1, activator protein-1; Apaf-1, apoptotic protease activating factor-1; AR, androgen receptor; ATF6, activating transcription factor-6; BAP31, B-cell receptor-associated protein 31; Bax, Bcl-2 associated X protein; Bcl-2, B-cell lymphoma-2; Bcl-xL, B-cell lymphoma extra-large; Bid, BH3- interacting-domain death agonist; CDCA, chenodeoxycholic acid; Cdk2, cyclin-dependent kinase 2; CHOP, C/EBP homologous protein; cIAP-1, cellular inhibitor of apoptosis 1; Cyt *c*, cytochrome *c*; DCA, deoxycholic acid; DR4, death receptor 4; Elk-1, ETS like-1 protein; ERα, estrogen receptor α; FasL, Fas ligand; FASN, fatty acid synthase; GADD45, growth arrest and DNA damage-inducible 45; GCDCA, glycochenodeoxycholic acid; IGFBP3, insulin-like growth factor binding protein 3; IRE1α, inositol-requiring transmembrane kinase endoribonuclease-1α; IκB-α, nuclear factor-kappa-B inhibitor alpha; LCA, lithocholic acid; LDH, lactate dehydrogenase; Mcl-1, myeloid cell leukemia-1; MMP-2, matrix metalloprotein-2; NF-κB, nuclear factor kappa B; p-ATF2, phosphorylation of activating transcription factor 2; PARP, poly(ADP-ribose) polymerase; p-ERK, phosphorylation of extracellular signal-regulated protein kinases; p-FAK, phosphorylation of focal adhesion kinase; p-JNK, phosphorylation of c-Jun N-terminal kinases; PKC protein kinase C; pRb, phosphorylation of retinoblastoma protein; p-MEK, phosphorylation of MAP kinase/ERK kinase; p-mTOR, phosphorylation of mammalian target of rapamycin; PUMA, p53 upregulated modulator of apoptosis; ROS, reactive oxygen species; SIRT-1, sirtuin-1; SREBP-1c, sterol regulatory element-binding protein-1c; TGF-β, transforming growth factor-β; TGR5, G protein-coupled bile acid receptor 1; TNF-R1, tumor necrosis factor receptor 1; TRAIL, tumor necrosis factor-related apoptosis-inducing ligand; UDCA, ursodeoxycholic acid; VEGF, vascular endothelial growth factor; XIAP, X-linked inhibitor of apoptosis protein; ΔΨm, mitochondrial membrane potential.

**Table 2 ijms-23-07184-t002:** Molecular targets of bile acid-induced autophagy.

Types	Target Molecules		Model (s)	Refs.
	Up-Regulation	Down-Regulation		
**DCA**	Beclin-1		Esophageal adenocarcinoma (CP-A)	[109]
**LCA**	p-JNK, p-eIF2α, CHOP, ROS, caspase-3, LC3BⅡ, ATG5	BIM, PUMA,	Prostate cancer cells (PC-3, DU-145)	[110]
**UDCA**	Cleaved caspase-3, Cyt *c*, cleavage of PARP, LC3Ⅱ, caspase-3, -6, and -8, DR5, c-FLIP(L)	ATG5	Gastric cancer subline (SNU601/WT, SNU601/R)	[111]

ATG5, autophagy related 5; BIM, B cell lymphoma-like protein 11; CHOP, C/EBP homologous protein; Cyt *c*, cytochrome *c*; c-FLIP(L), long splice variant of cellular FLICE-inhibitory proteins; DCA, deoxycholic acid; DR5, death receptor 5; FXR, farnesoid X receptor; GCDCA, glycochenodeoxycholic acid; p-AMPK, phosphorylation of AMP-activated protein kinase; p-eIF2α, phosphorylation of eukaryotic initiation factor 2-alpha; p-JNK, phosphorylation of c-Jun N-terminal kinases; p-MLKL, phosphorylation of mixed lineage kinase domain-like; PUMA, p53 upregulated modulator of apoptosis; LCA, lithocholic acid; LC3Ⅱ, microtubule-associated protein 1A/1B-light chain 3Ⅱ; mTOR, mammalian target of rapamycin; PARP, poly(ADP-ribose) polymerase; RIP3, receptor-interacting protein 3; ROS, reactive oxygen species; UDCA, ursodeoxycholic acid.

**Table 3 ijms-23-07184-t003:** Molecular targets of bile acid-induced necroptosis.

Types	Target Molecules		Model (s)	Refs.
	Up-Regulation	Down-Regulation		
**CA**	RIPK3, p-RIPK3, IL-8, p-JNK		Hepatocellular carcinoma cells (HepG2)	[152]
**TCA**	FXR	ATG7	Rat pancreatic acinar-like cancer cell line (AR42J)	[159]
	FXR, SQSTM1/p62, FOXO3 (cytosolic), MLKL, caspase-3, -8, and -9, Bax, RIPK3, p-MLKL	ATG5, ATG7, LC3 (LC3-II), Beclin-1	Chronic pancreatitis tissue	[159]
**UDCA**	RIPK3, p-RIPK3, p-JNK		Hepatocellular carcinoma cells (HepG2)	[152]
**CDCA**	p-RIPK3, IL-8		Hepatocellular carcinoma cells (HepG2)	[152]
**GCDCA**	MLKL, p-MLKL, RIPK3		Liver of patients with PBC	[155]
	MLKL, p-MLKL, RIPK1, RIPK3		Liver of mice after BDL	[155]
	FXR	ATG5, ATG7	Pancreatic cancer cell lines (MIA PaCa-2, BxPC-3)	[159]
	FXR	ATG5, ATG7	Rat pancreatic acinar-like cancer cell line (AR42J)	[159]
	FXR, SQSTM1/p62, FOXO3 (cytosolic), MLKL, caspase-3, -8, and -9, Bax, RIPK3, p-MLKL	ATG5, ATG7, LC3 (LC3-II), Beclin-1	Chronic pancreatitis tissue	[159]
**TCDCA**		RIPK3, p-RIPK3,	Hepatocellular carcinoma cells (HepG2)	[152]
**LCA** **GLCA** **TLCA**		RIPK3, p-RIPK3	Hepatocellular carcinoma cells (HepG2)	[152]

ATG5, autophagy related 5; Bax, Bcl-2 associated X protein; CA, cholic acid; CDCA, chenodeoxycholic acid; FOXO3, forkhead box O3; FXR, farnesoid X receptor; GCDCA, glycochenodeoxycholic acid; GLCA, glycolithocholic acid; IL-8, interleukin 8; p-JNK, phosphorylation of c-Jun N-terminal kinases; LCA, lithocholic acid; LC3, microtubule-associated protein 1A/1B-light chain 3; MLKL, mixed lineage kinase domain-like; RIPK3, receptor-interacting protein kinase 3; SQSTM1/p62, sequestosome 1; TCA, taurocholic acid; TLCA, taurolithocholic acid; UDCA, ursodeoxycholic acid.

**Table 4 ijms-23-07184-t004:** Molecular targets of synthetic bile acid derivative-induced programmed cell death.

Derivatives	Types	Mechanism	Target Molecules		Model (s)	Refs.
			Up-Regulation	Down-Regulation		
**HS-1030**	UDCA derivative	apoptosis	p21	cyclin E, Cdk2, Cdk4, Cdk6, E2F-1	Colon cancer cells (HT-29)	[22]
**HS-1183**	UDCA derivative	apoptosis	cleavage of PARP	pro-caspase-3 and -8	Leukemic T cells (Jurkat cells)	[161]
		apoptosis	p21	cyclin D1, Cdk4, Cdk6, E2F-1	Colon cancer cells (HT-29)	[22]
		apoptosis	cleavage of PARP, Bax, c-Jun, p-JNK, AP-1	p-p38, p50, p65, IkB-α	Cervical carcinoma cells (SiHa)	[162]
		apoptosis	Bax, cleavages of lamin B and PARP, p21, p53	Bcl-2, cyclin D3, pRb	Breast cancer cells (MCF-7, MDA-MB-231)	[163]
		apoptosis	p21	pRb	Prostate cancer cells (PC-3)	[164]
**HS-1199**	CDCA derivative	apoptosis	cleavage of PARP	pro-caspase-3 and -8	Leukemic T cells (Jurkat cells)	[161]
		apoptosis	p21	cyclin D1, cyclin E, Cdk2, Cdk4, Cdk6, E2F-1	Colon cancer cells (HT-29)	[22]
		apoptosis	cleavage of PARP, Bax, c-Jun, p-JNK, AP-1	p-p38, p-ERK	Cervical carcinoma cells (SiHa)	[162]
		apoptosis	Bax, cleavages of lamin B and PARP, p21, p53	Bcl-2, cyclin D1, cyclin D3, pRb	Breast cancer cells (MCF-7, MDA-MB-231)	[163]
		apoptosis	cleavage of PARP, p21	pRb, cyclin D1, cyclin D3	Prostate cancer cells (PC-3)	[164]
		apoptosis	Cyt *c*, cleavage of PARP and DFF45, AIF(N), Nur77	ΔΨm, pro-caspase-3, XIAP	Stomach cancer cells (SNU-1)	[165,166]
		apoptosis	cleavage of PARP, Cyt *c*	pro-caspase-3, ΔΨm	Malignant glioblastoma cells (U-118MG, U-87MG, T98G, U-373MG)	[167]
**HS-1200**	CDCA derivative	apoptosis	cleavage of PARP	pro-caspase-3 and -8	Leukemic T cells (Jurkat cells)	[161]
		apoptosis	cleavage of PARP, p21	cyclin D1, cyclin E, Cdk2, Cdk4, Cdk6, pRb, E2F-1	Colon cancer cells (HT-29)	[22]
		apoptosis	cleavage of PARP, Bax, c-Jun, p-JNK, AP-1	p-p38, p-ERK, p65(total), p50(total), IkB-α(total)	Cervical carcinoma cells (SiHa)	[162]
		apoptosis	Bax, cleavages of lamin B and PARP, p21, p53, AIF	Bcl-2, cyclin D1, cyclin D3, pRb	Breast cancer cells (MCF-7, MDA-MB-231)	[163,168]
		apoptosis	cleavage of PARP, p21	pRb, Cdk2, cyclin D1, cyclin D3	Prostate cancer cells (PC-3)	[164]
		apoptosis	Cyt *c*, cleavage of PARP and DFF45, AIF(N), Nur77	ΔΨm, pro-caspase-3, XIAP	Stomach cancer cells (SNU-1)	[165,166]
		apoptosis	cleavage of PARP, Cyt *c*	pro-caspase-3, ΔΨm	Malignant glioblastoma cells (U-118MG, U-87MG, T98G, U-373MG), Xenografts (U87MG)	[167]
		apoptosis	Bax, p53, p21, p27, Egr-1, caspase-3 and 9, cleavage of PARP, Cyt *c* (cytosolic)	Bcl-2, cyclin D1, Cdk2, cyclin A, E2F-1, Mdm2, COX-2, Cyt *c* (mitochondrial), ΔΨm	Hepatoma cells (HepG2, Hep3B, BEL7402)	[169,170]
		apoptosis	AIF, CAD, cleavage of PARP	ΔΨm, pro-caspase-3 and -7, PARP	Thyroid carcinoma (KAT 18)	[171]
**ent-CDCA**	enantiomers of CDCA	apoptosis		pro-caspase-3 and -9	Colon cancer cells (HT-29, HCT-116)	[23]
**ent-DCA**	enantiomers of DCA	apoptosis		pro-caspase-3 and -9	Colon cancer cells (HT-29, HCT-116)	[23]
**ent-LCA**	enantiomers of LCA	apoptosis	CD95, ROS,	pro-caspase-2, -3, -8, and -9, Bid	Colon cancer cells (HT-29, HCT-116)	[23]
**6af and 6cf**	bile-acid-appended triazolyl aryl ketones	apoptosis			Breast cancer cells (MCF-7)	[24]
**CDC-PTX** **UDC-PTX**	CDCA derivative UDCA derivative	apoptosis			Acute promyelocytic leukemia cells (HL60, NB4)	[25]
	CDCA derivative UDCA derivative	apoptosis			Colon cancer cells (RKO, HCT-116)	[25]
**7b**	Piperazinyl bile acid derivative	apoptosis			Multiple myeloma (KMS-11), Colonic cancer cells (HCT-116)	[26]
**LCA-TMA_1_, CDCA-TMA_2_, DCA-TMA_2_, and CA-TMA_3_**	cationic bile acid based facial amphiphiles featuring trimethyl ammonium head groups	apoptosis			Colon cancer cells (HCT-116, DLD-1)	[27]
**compound IIIb**	CDCA-substituted piperazine conjugate	apoptosis	cleavage of Mcl-1 and PARP-1, Ip-IκBα, DNA fragmentation	IκBα	Multiple myeloma (KMS-11)	[28]
**LCA-PIP1**	LCA amphiphile	apoptosis		pro-caspase-3, -7, and -8	Colon cancer cells (HCT-116) Xenograft (HCT-116)	[29]
**CA-Tam_3_-Am**	CA−tamoxifen conjugate	apoptosis	Bax, Bid, Bad, caspase-9, cleaved caspase-3 and -8, Cyt *c*, ROS	Bcl-2, Bcl-xL, survivin	Breast cancer cells (MCF-7, T47D, MDA-MB 231)	[30]
**norUDCA**	UDCA derivative	autophagy	ratio of LC3-II to LC3-I, ATG5, ATG5/ATG12, p-AMPK, p-ULK1(Ser317), p-ULK1(Ser555), p-ULK1(Ser777)	p62, α1ATZ, p-mTOR, p-ULK1(Ser757)	Cervical cancer cells (HTOZ)	[31]
**compound 9**	DCA derivative	apoptosis, autophagy	caspase-3 and -7, ROS		Duodenal carcinoma cells (HuTu 80)	[32]

AIF, apoptosis-inducing factor; AP-1, activator protein-1; ATG5, autophagy related 5; BAD, Bcl-2 antagonist of cell death; Bax, Bcl-2 associated X protein; Bcl-2, B-cell lymphoma-2; Bcl-xL, B-cell lymphoma extra-large; Bid, BH3-interacting-domain death agonist; CA, cholic acid; CAD, caspase-activated DNase; CA-TMA_3_, cholic acid based amphiphile; CA-Tam_3_-Am, cholic acid−tamoxifen conjugate; CDCA, chenodeoxycholic acid; CDC-PTX, chenodeoxycholic-paclitaxel hybrid; CDCA-TMA_2_, chenodeoxycholic acid based amphiphiles; Cdk2, cyclin-dependent kinase 2; CD95, cluster of differentiation 95; compound IIIb, chenodeoxycholic acid-substituted piperazine conjugate; compound 9, chenodeoxycholic acid derivative; COX-2, cyclooxygenase-2; Cyt *c*, cytochrome *c*; DCA, deoxycholic acid; DCA-TMA_2_, deoxycholic acid based amphiphiles; DFF45, DNA fragmentation factors 45; Egr-1, early growth response-1; ent-CDCA, enantiomers of chenodeoxycholic acid; ent-DCA, enantiomers of deoxycholic acid; ent-LCA, enantiomers of lithocholic acid; E2F1, E2 promoter binding factor 1; HS-1030 and HS-1183, ursodeoxycholic acid derivatives; HS-1199 and HS-1200, chenodeoxycholic acid derivatives; IκBα, nuclear factor of kappa light polypeptide gene enhancer in B-cells inhibitor, alpha; LCA, lithocholic acid; LCA-PIP_1_, lithocholic acid–piperidine; LCA-TMA_1_, lithocholic acid based amphiphile; norUDCA, nor-ursodeoxycholic acid; LC3Ⅱ, microtubule-associated protein 1A/1B-light chain 3Ⅱ; Mcl-1, myeloid leukemia 1; p-AMPK, phosphorylation of AMP-activated protein kinase; PARP, poly(ADP-ribose) polymerase; p-ERK, phosphorylation of extracellular signal-regulated protein kinases; p-JNK, phosphorylation of c-Jun N-terminal kinase; p-mTOR, phosphorylation of mTOR, mammalian target of rapamycin; pRb, phosphorylation of retinoblastoma; p-ULK1, phosphorylation of Unc-51 like autophagy activating kinase 1; ROS, reactive oxygen species; UDCA, ursodeoxycholic acid; UDC-PTX, ursodeoxycholic-paclitaxel hybrid; XIAP, X-linked inhibitor of apoptosis protein; α1ATZ, α1AT mutant Z; ΔΨm, mitochondrial membrane potential; 6af and 6cf; bile acid-added triazolyl aryl ketones; 7b, piperazinyl bile acid derivative.

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
