# Peer review of "Mechanism of Bile Acid-Induced Programmed Cell Death and Drug Discovery against Cancer: A Review"

_ijms, 2022, doi:10.3390/ijms23137184_

Round 1

Reviewer 1 Report

Bile acids, the major constituents of bile that play crucial biological roles, such as solubilization of lipids in the intestinal lumen, among many others. In this manuscript, the authors have reviewed the role of bile acid in programmed cell death, such as apoptosis, autophagy, and necroptosis, and the status of drug development using synthetic bile acid derivatives against human cancers.

Minor revision

Comment1: The strength of the manuscript is that it covers most of the aspects of bile acid induced programmed cell death and inform the status of Drug development using synthetic bile acid derivatives, yet the manuscript is concise. Weaknesses include some grammatical errors and organization of some subsections. Also, while the conclusions section suggests areas of research that are ripe for investigation, the authors do not contribute any new hypotheses.

Comment 2: There are some recent articles regarding the effect of excessive bile acids on autophagy (e. g. https://doi.org/10.1002/jcp.30774). The author should include that in the manuscript.

Comment 3: For bile acid induced necrosis the authors should cite all the relevant documents. They should incorporate this citation (Toxicol Appl Pharmacol. 2015 Mar 15; 283(3): 168–177.)

Comment 4: The section “Cell death Mechanism of Synthetic Bile acid derivatives in Cancer” can be improved. Specially Figure 3 can be more compact. The review should provide the reader with an analysis and interpretation of published work. The authors can summarize others’ findings, make the respective section brief and to the point.

Comment 5: line 42: It should be ‘improve’ not ‘improves’

Comment 6: The authors can include a diagram for bile acid induced programmed cell death.

Author Response

  • Point 1: The strength of the manuscript is that it covers most of the aspects of bile acid induced programmed cell death and inform the status of Drug development using synthetic bile acid derivatives, yet the manuscript is concise. Weaknesses include some grammatical errors and organization of some subsections. Also, while the conclusions section suggests areas of research that are ripe for investigation, the authors do not contribute any new hypotheses.

Response 1: Thanks for your comment. We proposed at the end of our discussion that new bile acid derivatives may be promising chemicals that specifically target a variety of cancer cells by inducing programmed cell death. In other words, based on the structure of bile acids, it suggests that bile acid derivatives can be important lead compounds for the development of new anticancer drugs.

  • Point 2: There are some recent articles regarding the effect of excessive bile acids on autophagy (e. g. https://doi.org/10.1002/jcp.30774). The author should include that in the manuscript.

Response 2: Thanks for your comment. It was helpful to point out the most recent paper on bile acids that we were unaware of. The paper you mentioned (https://doi.org/10.1002/jcp.30774 = J Cell Physiol. 2022;1–15) states that the complex interaction of excessive bile acids induced autophagy flux, mitochondrial dysfunction, and and cellular apoptosis in placental trophoblasts may play a critical role in the pathogenesis of intrahepatic cholestasis of pregnancy. On the other hand, our paper discusses bile acid-induced programmed cell death (apoptosis, autophagy, and necroptosis) in a model of human cancer. However, the necroptosis is a recent theory of programmed cell death and has not been studied much. Therefore, only necroptosis  was reviewed for cholestasis and bile acids associated with pancreatitis in addition to human cancer. And since the autophagy part of our review is from the content of the human cancer model only, we have decided not to add the article you suggested, so please understand.

  • Point 3: For bile acid induced necrosis the authors should cite all the relevant documents. They should incorporate this citation (Toxicol Appl Pharmacol. 2015 Mar 15; 283(3): 168–177.).

Response 3: Thanks for your advice. Our reivew is a discussion of bile acid-induced programmed cell death. Your proposed paper is about bile acid-induced necrosis in primary human hepatocytes and in patients with obstructive cholestasis. Necrosis is not involved in programmed cell death. Therefore, we have decided not to add the article you suggested, so please understand.

  • Point 4: The section “Cell death Mechanism of Synthetic Bile acid derivatives in Cancer” can be improved. Specially Figure 3 can be more compact. The review should provide the reader with an analysis and interpretation of published work. The authors can summarize others’ findings, make the respective section brief and to the point.

Response 4: Thanks. We have modified Figure 3 to be more concise based on your comments on the manuscript.

  • Point 5: line 42: It should be ‘improve’ not ‘improves’

Response 5: We are very sorry for the mistake. We edited the manuscript.

  • Point 6: The authors can include a diagram for bile acid induced programmed cell death.

Response 6: Thanks for your suggestion. We have tabulated the bile acid-induced programmed cell death (apoptosis, autophagy, and necroptosis) and synthetic bile acid-induced programmed cell death. Next time, we will consider writing including diagram.

Reviewer 2 Report

In this review, Jung Yoon Jang and colleagues summarize studies describing the role of bile acids in different types of cell death and how they can be used as anticancer drugs.

The authors generally describe the effect of bile acids on cancer and then explain the types of cell death and how bile acids influence them.

Abstract: the authors should mention that bile acids are amphipathic molecules involved in important physiological processes as intestinal absorption of nutrients as well as the secretion of toxins and xenobiotics in liver (a detoxifying organ). BA are signaling molecules but it is important to mention the digestive, metabolic regulators and detoxifying functions of these molecules in the abstract, as well as mention what kind of molecules.

As metabolic regulators they activate different signaling pathways involved in the development and progression of different diseases.

Introduction: authors should describe how and where bile acids are synthesized, e.g. the enzymes involved and where these enzymes are, the transporters involved, etc. As this review focuses on bile acids, it is important to describe them in more detail. This fact is briefly explained on line 28.

I think that in the role of BA in cancer (line 87) the authors focus too much on in vitro studies. The rationale for the review would be better understood if the effect of BAs on different types of cancer in humans were explained in more detail at this point, and then confirmed in vitro. Similarly, in the other sections related to cell death, the authors focus more on in vitro studies. I think explaining in more detail what happens in humans would strengthen this review as it should be the starting point of any original article or review. .

Minor:

Figure 2: Consider placing the figure and legend on another page as it is difficult to find the end of the legend this way. Line 54 enterohepatic circulation instead of enterohepatic recirculation.

Line 139 Environmental factors inside and outside the body sound strange. I would delete it.

Line 141: I would include alcohol as it is metabolized in the liver and produce cholestasis in patients with alcoholic liver disease.

Author Response

  • Point 1: Abstract:the authors should mention that bile acids are amphipathic molecules involved in important physiological processes as intestinal absorption of nutrients as well as the secretion of toxins and xenobiotics in liver (a detoxifying organ). BA are signaling molecules but it is important to mention the digestive, metabolic regulators and detoxifying functions of these molecules in the abstract, as well as mention what kind of molecules.

As metabolic regulators they activate different signaling pathways involved in the development and progression of different diseases.

Response 1: Thanks for your comment. We have added your suggestions to the abstract and modified them accordingly.

  • Point 2: Introduction: authors should describe how and where bile acids are synthesized, e.g. the enzymes involved and where these enzymes are, the transporters involved, etc. As this review focuses on bile acids, it is important to describe them in more detail. This fact is briefly explained on line 28.

I think that in the role of BA in cancer (line 87) the authors focus too much on in vitro studies. The rationale for the review would be better understood if the effect of BAs on different types of cancer in humans were explained in more detail at this point, and then confirmed in vitro. Similarly, in the other sections related to cell death, the authors focus more on in vitro studies. I think explaining in more detail what happens in humans would strengthen this review as it should be the starting point of any original article or review.

Response 2: Thanks for your comment. We’ve added some information about the role of bile acids in the human body, as you suggested. The added contents are as follows. In particular, what role UDCA played in the occurrence of various diseases, including colorectal cancer, was added to from Line 138 to Line 142. In addition, detailed descriptions of how primary bile acids are produced in the human liver from line 222 to line 237 have been made.

  • Point 3: Figure 2: Consider placing the figure and legend on another page as it is difficult to find the end of the legend this way. Line 54 enterohepatic circulation instead of enterohepatic recirculation.

Response 3: Thanks for your advice. We have arranged Figure 2 and the legend on the next page in the manuscript. In addition, it has been modified to enterohepatic circulation instead of enterohepatic recirculation.

  • Point 4: Line 139 Environmental factors inside and outside the body sound strange. I would delete it.

Response 4: Thanks for your suggestion. We have revised the manuscript accordingly.

  • Point 5: Line 141: I would include alcohol as it is metabolized in the liver and produce cholestasis in patients with alcoholic liver disease.

Response 5: Thanks for your suggestion. We revised the manuscript by adding 'alcohol' to your comment.

Round 2

Reviewer 2 Report

The authors have considered and included the suggested comments. I think this review deserves to be published.